# Unexpected predicted length variation for the coding sequence of the sleep related gene, *BHLHE41* in gorilla amidst strong purifying selection across mammals

Krishna Unadkat, Justen B. Whittall◉*

Department of Biology, Santa Clara University, Santa Clara, California, United States of America

* jwhittall@scu.edu

**Data Availability Statement:** All relevant data are within the paper and its Supporting Information files.

## Abstract

There is a molecular basis for many sleep patterns and disorders involving circadian clock genes. In humans, "short-sleeper" behavior has been linked to specific amino acid substitutions in *BHLHE41* (DEC2), yet little is known about variation at these sites and across this gene in mammals. We compare *BHLHE41* coding sequences for 27 mammals. Approximately half of the coding sequence was invariable at the nucleotide level and close to three-quarters of the amino acid alignment was identical. No other mammals had the same "short-sleeper" amino acid substitutions previously described from humans. Phylogenetic analyses based on the nucleotides of the coding sequence alignment are consistent with established mammalian relationships confirming orthology among the sampled sequences. Significant purifying selection was detected in about two-thirds of the variable codons and no codons exhibited significant signs of positive selection. Unexpectedly, the gorilla *BHLHE41* sequence has a 318 bp insertion at the 5' end of the coding sequence and a deletion of 195 bp near the 3' end of the coding sequence (including the two short sleeper variable sites). Given the strong signal of purifying selection across this gene, phylogenetic congruence with expected relationships and generally conserved function among mammals investigated thus far, we suggest the indels predicted in the gorilla *BHLHE41* may represent an annotation error and warrant experimental validation.

## Introduction

Sleep plays a vital function for survival in animals [1–3], especially vertebrates and even some invertebrates [4]. It is essential in maintaining both physical and mental health, especially in humans where sleep deprivation is linked to diabetes, high blood pressure, obesity, and decreased immune function [5,6,7]. The timing and duration of sleep varies widely among mammals [8] and is regulated by a plethora of intricate mechanisms including many circadian clock genes [9].

Among the genes responsible for circadian regulation in mammals is the basic helix-loop-helix family member e41 [5, 10, 11], also known as "differentially expressed in chondrocytes

**Funding:** The author(s) received no specific funding for this work.

**Competing interests:** The authors have declared that no competing interests exist.

protein 2" (*DEC2*). It is an essential clock protein that acts as a transcription factor which maintains the negative feedback loop in the circadian clock by repressing E-box-mediated transcription [5]. Specifically, by binding to the promoter region on the *prepro-orexin* gene, *BHLHE41* acts as a repressor of orexin expression in mammals. Furthermore, disabling orexin results in narcolepsy in mammals, confirming that orexin plays a vital role in sleep regulation [5].

*BHLHE41* has several conserved functional domains including a bHLH region and the "orange" domain. As a member of the bHLH family, *BHLHE41* contains a ~60 amino acid bHLH conserved domain that promotes dimerization and DNA binding [10]. Specifically, the bHLH domain is composed of a DNA-binding region, E-box/N-box specificity site, and a dimerization interface for polypeptide binding. The DNA-binding region is followed by two alpha-helices surrounding a variable loop region. As a member of the group E bHLH family, this protein specifically binds to an N-box sequence (CACGCG or CACGAG) based on *BHLHE41* amino acid site 53 (glutamate) [12]. The other well studied conserved domain in *BHLHE41* is the orange domain which provides specificity as a transcriptional repressor [13]. These domains are conserved between humans and zebrafish in both their amino acid composition and function [14]. Unfortunately, there is no 3D structure described for a mammalian *BHLHE41* in Genbank's Protein Data Bank [15] to determine the spatial effects of amino acid variants.

Because of its essential function in sleep regulation, anomalies in clock genes can lead to abnormal patterns of sleep that can manifest in a wide variety of ways, ranging from insomnia to oversleeping [1]. A rare point mutation in the *BHLHE41* gene of *Homo sapiens* (P384R in NM_030762, also referred to as P385R as in [10]) confers a "short-sleeper phenotype". The mutation involves a transversion from a C to G in the DNA sequence of *BHLHE41*, which results in a non-synonymous substitution from proline to arginine at amino acid position 385 of the *BHLHE41* protein. Since proline (nonpolar) and arginine (electrically charged, basic) have chemically dissimilar structures and since substituting these amino acids is relatively rare (BLOSUM62 value of -2), it is not surprising that this mutation has a substantial phenotypic effect. Subjects with this allele reported shorter daily sleep patterns than those with the wild type allele, without reporting any other adverse effects [10]. The function of *BHLHE41* in controlling sleep and circadian clocks is conserved between humans and mice, but untested in most other mammals [10]. In zebrafish, the *BHLHE41* has similar structure (five exons separated by four introns) and high sequence similarity to human homologue [14], but no variation at this residue. In *Drosophila melanogaster*, the most similar gene to *BHLHE41* is *CG17100* (*Clockwork Orange*), but is only weakly similar (<11% amino acid identity; [16]). However, transgenically introducing the short-sleeper allele P385R into *Drosophila* still resulted in the short-sleeper phenotype [10] suggesting the existence of a similar regulatory network. Another nonsynonymous substitution in *BHLHE41* that correlates with altered sleep behavior in humans is Y362H [17]. This mutation reduced the ability of *BHLHE41* to suppress *CLOCK/BMAL1* and *NPAS2/BMAL1* transactivation in vitro [17].

These short-sleeper variants could provide adaptive functions in other mammals. In such case, we may detect the signature of positive selection on those codons. However, genes such as *BHLHE41* are essential for survival and reproduction and are therefore often highly conserved and are more likely to show patterns of purifying selection. Purifying selection can be manifested as higher rates of synonymous substitutions compared to rates of non-synonymous substitutions (dN-dS) [18]. Negative overall dN-dS values indicate purifying selection and are often evidence that a gene is involved in some essential function (like the circadian clock), yet a codon-by-codon dN/dS analysis can detect signs of positive selection (*e.g*,. adaptation at the molecular level) on specific codons. To date, no one has examined patterns of selection in *BHLHE41*.

In fact, very few nucleotide, nor amino acid comparisons have been made in mammals beyond human vs. mouse. With the rapid accumulation of mammalian genome sequences, a plethora of homologous sequences likely exist (see [12] for phylogenetic analysis of all bHLH, but only includes two mammals—human and mouse; see [14] for a comparison of zebrafish and human that calls for further sampling of mammals). Furthermore, the well-resolved mammalian phylogeny [19, 20] provides a robust foundation for which to test for homology and confirm orthology. For most non-model mammalian species with whole-genome sequences, genes are predicted using algorithms that locate open reading frames (*e.g.*, [21]), yet rarely are the predicted genes validated experimentally [22, 23]. Some algorithms compare putative open reading frames with model-species to confirm length and expected sequence variation. Accounting for any differences in the length of coding sequences can be a challenge, due to both the existence of alternative mRNA isoforms and an increasing time of divergence [24]. A comparative approach across a diversity of lineages can help elucidate any unusual patterns of sequence variation.

In order to further explore the function of the *BHLHE41* gene, we analyzed the evolutionary relationships among the *BHLHE41* coding sequence in humans and other mammals. There are two clear aims of this study: (1) to utilize pre-existing data in Genbank to determine whether any mammals other than humans have the "short-sleeper" allele or exhibit variation at amino acid sites P385R and Y362H, and (2) to assess the degree of biochemical changes at all amino acid substitutions and search for the footprints of selection (dN-dS). To address these goals, we compared *BHLHE41* sequences from 27 species of mammals and a reptilian outgroup that came from sequenced cDNA and full genome sequencing projects. After creating a multiple sequence alignment, we used Bayesian and maximum likelihood analyses to investigate the evolutionary relationships underlying this gene among mammals to confirm orthology. Finally, we used the multiple sequence alignment to test for purifying and positive selection across codons.

## Materials and methods

### Query sequence search

In order to find the complete mRNA coding sequence for *BHLHE41* from *H. sapiens*, we searched ENTREZ using the "RefSeq" filter with the following query to the "Gene" database: "DEC2 AND *H. sapiens* [organism]". We confirmed that the same sequence was obtained when searching for "*BHLHE41* AND *H. sapiens* [organism]". We found a single hit for the *Homo sapiens BHLHE41* gene with the RefSeq accession number NM_030762 [25]. The coding sequence for this gene is 1449 base pairs long. According to EMBL (ENSGGOT00000015550.3), there are five introns (yet see [14] where they report only four introns). All subsequent analyses are based solely on the coding sequence as determined by EMBL.

### Locating homologous sequences with BLAST

After locating the accession number for our sequence of interest from *H. sapiens*, we used NCBI's nucleotide BLAST [26] to find other mammalian homologues to the *H. sapiens BHLHE41* mRNA. We searched the nucleotide collection (nr/nt) using Megablast with default parameters (Word Size: 28, Match/Mismatch: 1, -2, Gap Costs: Linear). We downloaded sequences with E-values $< 10^{-3}$, local percent identity $> 70\%$, and query coverages ~100% as Genbank complete flatfiles.

In order to find an outgroup sequence, we performed another BLAST search using Discontiguous Megablast with default parameters (Word size: 1, Match/Mismatch: 2, -3, Gap Existence/Extension Costs: 5, 2) except excluding mammals from the search results. We included

the reptile, *Pelodiscus sinensis* or Chinese Softshell Turtle, as our outgroup based on the afore-mentioned E-value, identity, and query coverage cut-offs. GenBank flatfiles for each species coding sequence was downloaded and imported into Geneious Prime (Biomatters, New Zealand).

## Multiple sequence alignment

In order to create an alignment with sequences that represent homology to the *H. sapiens* BHLHE4 mRNA, we used the Geneious Aligner within Geneious Prime. To prevent single nucleotide gaps and ensure all remaining nucleotide gaps were in multiples of three, since this is coding sequence, we applied a cost matrix of 70% similarity (match/mismatch of 5.0/-4.5), a gap open penalty of 90, a gap extension penalty of one, and two refinement iterations.

## Phylogenetic analyses

In order to test for homology and confirm we were comparing orthologous sequences, we conducted maximum likelihood and Bayesian phylogenetic analyses. If the evolutionary relationships of the *BHLHE41* coding sequence reflects the known relationships among mammals, then we can conclude homology and proceed with the tests for selection. In order to construct a maximum likelihood tree, we used the RAxML v.4.0 [27] plugin in Geneious Prime. We applied the GTR+CAT+I model of sequence evolution, with the Rapid Bootstrapping algorithm, 1,000 bootstrap replicates, and a Parsimony Random Seed of one. This is the most complex model of sequence evolution available for the RAxML plugin in Geneious Prime. It accounts for six rates of nucleotide substitution with categories for rate variation instead of a gamma distribution for efficiency, while simultaneously estimating the proportion of invariant sites [27].

To compare our maximum likelihood results with another method, we constructed a phylogenetic tree using the MrBayes v.2.2.4 [28] plugin from within Geneious Prime. For this analysis, we used the GTR (General Time Reversible) model of sequence evolution with "gamma" rate variation. The search ran for 2,000,000 generations, subsampling every 1,000 generations after 1,000,000 generations of burnin. Two parallel runs were conducted using four chains each with a heated chain temp of 0.2. In order to confirm sufficient number of generations were sampled in the Bayesian analysis, we recorded the standard deviation of split frequencies comparing the two runs. Furthermore, we examined the trace depicting the maximum likelihood value at each generation to ensure there was no slope (S1 Fig). After both maximum likelihood and Bayesian trees were generated, we rooted them with the reptilian outgroup, *P. sinensis* (Chinese Softshell Turtle).

## Molecular evolution

By comparing the rates of nonsynonymous (dN) and synonymous (dS) substitutions, we tested for selection at the molecular scale. In MEGA7 [29], we used the codon-based Z-test of selection to test for pairwise dN-dS values, using "In Sequence Pairs" as the scope, "Positive Selection" as the test hypothesis, the "Nei-Gojobori method (Proportional)" as the model [30], and "Pairwise Deletion" to account for gaps without removing sites entirely. We then repeated this process using "Purifying Selection" as the test hypothesis. Purifying selection was represented by negative dN-dS values, positive selection was represented by positive dN-dS values. dN-dS values of zero represent neutrality. For the codon-based Z-test of selection, p-values under 0.05 were considered significant.

In order to determine if there was directional selection on any specific codons, we used HyPhy [31] from within MEGA7. We used a "Neighbor-Joining tree", "Maximum Likelihood"

statistical method, "Syn-Nonsynonymous" substitution, and the "General Time Reversible" model of sequence evolution to analyze the alignment codon-by-codon. We applied the partial deletion option if <70% of the sequences had a gap. After running HyPhy, we removed invariant codons where dN and dS could not be calculated and examined the remaining codons with significant P-values. Values greater than 0.95 were considered significant evidence of purifying selection. We estimated the average dN-dS values for both conserved domains compared to the remaining codons outside the conserved domains.

## Results

### Locating homologous sequences with BLAST

From the results of the BLAST search using *BHLHE41* from *H. sapiens*, we downloaded 27 mammalian sequences (including the query) and one reptile sequence as an outgroup for a total alignment of 28 species.The E-values for all sequences were 0.0 and the local identity scores from the BLAST report ranged from 87.50% to 100% (Table 1). The coding sequences ranged in length from 1368 (*P. sinensis*) to 1569 (*Gorilla gorilla gorilla*) base pairs. The query coverage from the BLAST report ranged from 38% to 100% (Table 1).

### Multiple sequence alignment

All sequences in the multiple sequence alignment are complete from start codon (AUG) to stop codon (all use TGA) (S3 Table). Indels ranged from three base pairs to 318 base pairs—always in multiples of three. Of the 1,794 bp nucleotide alignment for mammals, 986 bp were identical (55.0%). The average nucleotide pairwise identity among the mammalian sequences was 92.2%. At the amino acid level, of the 598 residues for mammals, 71.7% were identical. The pairwise percent identity in amino acids was 94.8% (S4 Table).

There are no amino acid substitutions in our alignment at either residue previously described to confer alternative sleep behaviors in humans (Y362H and P385R, site numbers refer to human sequence). In our multiple sequence alignment, Y362H is at amino acid alignment position 476 (S4 Table) and nucleotide alignment positions 1423-1425bp (S3 Table). There is also no nucleotide variation for this codon. Alternatively, P385R is at amino acid alignment position 498 (S4 Table) and nucleotide alignment positions 1489-1491bp (S3 Table). Although there are no amino acid substitutions at this residue, there is synonymous variation. All but four sequences have the codon CCG, which codes for proline. The exceptions are synonymous substitutions in *Sus scrofa* (CCA), *Rousettus aegyptiacus* (CCC), and *P. sinensis* (CCC)—all of which still code for proline. However, in the *G. gorilla gorilla* sequence, both residues 362 and 385 fall within the 195 base pair deletion described above.

### Sequence length variation in gorilla

There are two large indels in the gorilla sequence (Fig 1; S3 and S4 Tables). The first 318 base pairs are only present in accession XM_031000846.1—a predicted protein from the *G. gorilla gorilla* genome sequence [32]. Additionally, the sequence for *G. gorilla gorilla* has a 195 base pair deletion starting at nucleotide alignment site 1,360 and ending at 1,555bp. Both of these indels are multiples of three and therefore maintain the reading frame throughout the coding sequence yielding a predicted *G. gorilla gorilla BHLHE41* amino acid sequence 522 residues. The average non-gorilla mammalian amino acid sequence is 482aa long.

We searched the *Gorilla gorilla gorilla* chromosome 12 whole genome shotgun sequence (NC_018436) between bp 58,885,949 and 58,889,015 and found that although the unusual 318bp upstream from the mammalian start codon exists, the gorilla annotation actually

**Table 1. Sequences used in creating the multiple sequence alignment and their BLAST scores using the mRNA from the human basic helix-loop-helix family member e41 as the query.**

| Latin name | Accession number | Query coverage (%) | Identity (%) |
|---|---|---|---|
| *Bos indicus x Bos taurus* | XM_027541573 | 64 | 91.97 |
| *Callorhinus ursinus* | XM_025879601 | 82 | 92.94 |
| *Cebus capucinus* | XM_017507035 | 94 | 96.60 |
| *Cercocebus atys* | XM_012093655 | 46 | 98.21 |
| *Chlorocebus sabaeus* | XM_007967990 | 99 | 97.96 |
| *Delphinapterus leucas* | XM_022577811 | 95 | 87.54 |
| *Homo sapiens*[1] | NM_030762 | 100 | 100 |
| *Gorilla gorilla gorilla* | XM_031000846.1 | 89 | 98.92 |
| *Lagenorhynchus obliquidens* | XM_027129408 | 89 | 87.27 |
| *Lipotes vexillifer* | XM_007446307 | 38 | 93.63 |
| *Macaca fascicularis* | XM_005570417 | 100 | 98.25 |
| *Macaca mulatta* | XM_015151321 | 49 | 98.13 |
| *Macaca nemestrina* | XM_011759130 | 100 | 98.08 |
| *Marmota flaviventris* | XM_027934162 | 52 | 92.03 |
| *Microcebus murinus* | XM_012739537 | 93 | 93.71 |
| *Orcinus orca* | XM_004270956 | 51 | 89.93 |
| *Ovis aries* | XM_015093964 | 67 | 92.73 |
| *Panthera pardus* | XM_019452268 | 60 | 92.65 |
| *Pan troglodytes* | XM_520805 | 49 | 99.15 |
| *Pelodiscus sinesis* [2] | XM_006127674 | 49 | 88.53 |
| *Physeter catodon* | XM_024128992 | 85 | 87.50 |
| *Piliocolobus tephrosceles* | XM_023209042 | 100 | 97.17 |
| *Pongo abelii* | XM_002823045 | 48 | 98.98 |
| *Rousettus aegyptiacus* | XM_016119294 | 92 | 91.98 |
| *Sus scrofa* | XM_003355541 | 79 | 92.49 |
| *Theropithecus gelada* | XM_025402281 | 94 | 97.23 |
| *Tursiops truncatus* | XM_019936346 | 95 | 87.54 |
| *Zalophus californianus* | XM_027593397 | 87 | 92.89 |

[1] Query sequence

[2] Reptile outgroup

identified the correct start codon (no 318bp insertion on the 5' end). Yet, regarding the 195bp deletion near the 3' end, we found 224 N's between exon 5 and exon 6 which likely includes both intron 5 and the missing 195bps of exon 6. In this case, the gorilla annotation is clearly different from the predicted mRNA.

## Phylogenetic analyses

Both maximum likelihood and Bayesian phylogenetic analyses were highly congruent. There were 20 significantly supported branches in both the maximum likelihood phylogenetic analysis (Fig 2) and in the Bayesian phylogenetic analysis (S2 Fig). In both trees, *H. sapiens* and *Pan troglodytes* are strongly supported sister species (bootstrap = 97%, posterior probability = 0.99). Additionally, the Great Apes are monophyletic in both phylogenetic analyses. While the two trees support the same evolutionary relationships, they have one minor differences in terms of support values. In the tree generated using Bayesian analysis, two of the Old World monkeys (*Cercocebus atys* and *Theropithecus gelada*) are sister species with a strong posterior probability

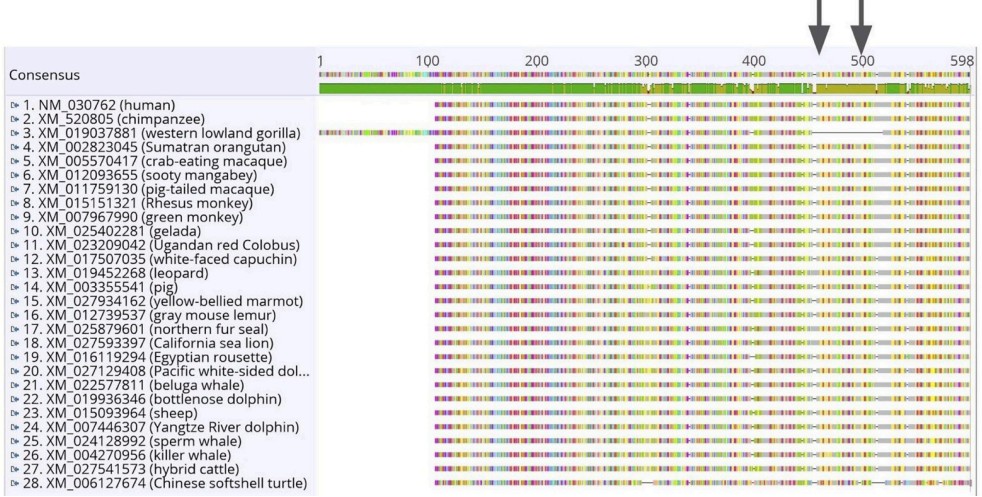

**Fig 1. Multiple sequence alignment of the *BHLHE41* mRNA for 27 mammals and one reptile outgroup.** Sequence identity is shown immediately below the consensus (green = 100% identical; gold = 25–99% identical; red < 25% identical). The two amino acid variants known to affect sleep behavior in humans (P385R and Y362H) are indicated with arrows. The alignment shows two unexpected findings in the gorilla sequence: a 318 base pair insertion on the 5' end and a 195 base pair gap starting at bp 1360.

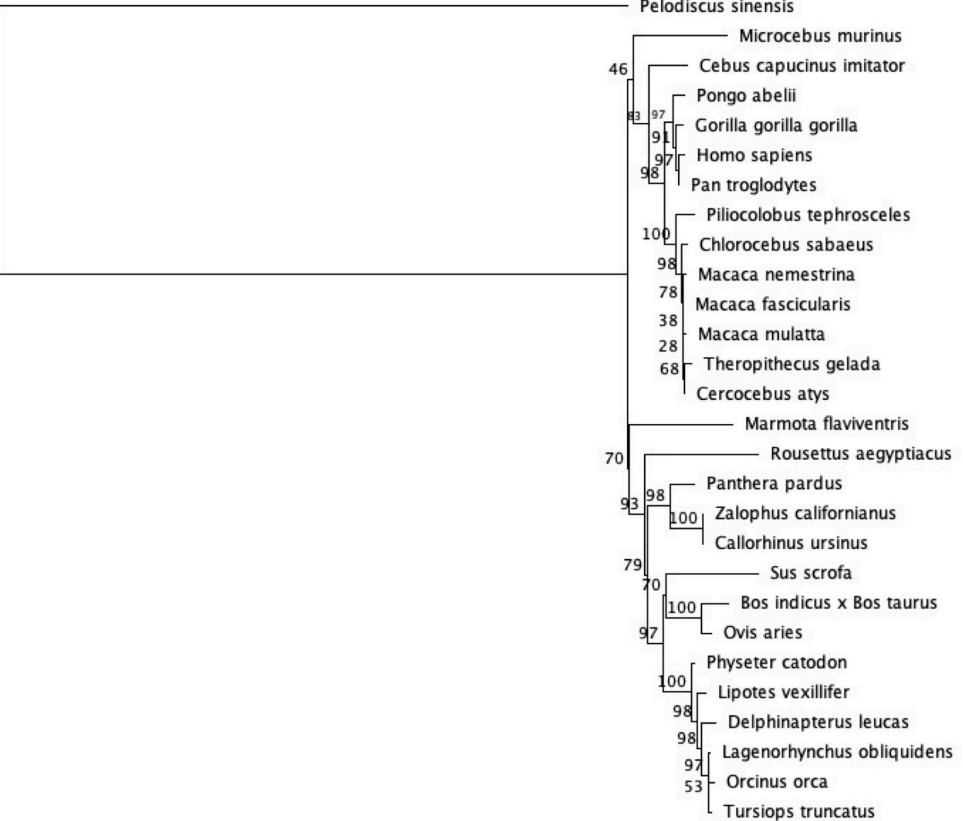

**Fig 2. Maximum likelihood phylogenetic analysis of mammalian *BHLHE41* coding sequence.** We used the GTR +CAT+I parameter settings with 100 bootstrap replicates which are indicated next to the branches. The tree is rooted with the reptilian outgroup, *Pelodiscus sinensis*.

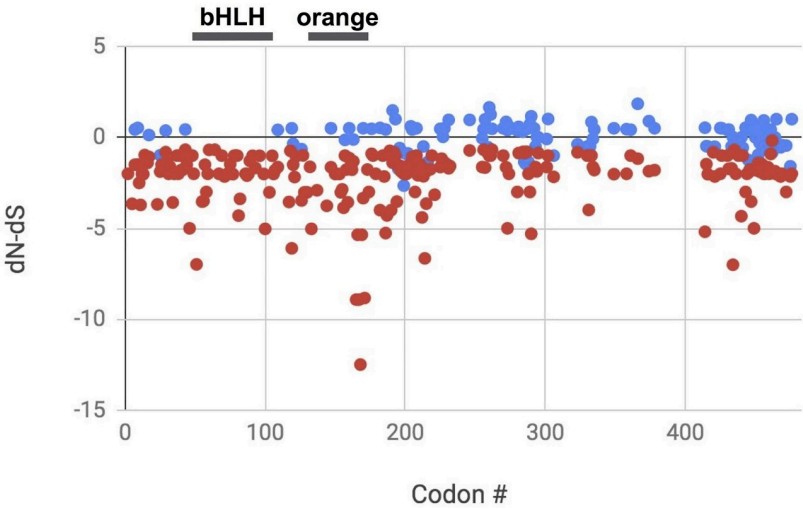

**Fig 3. Codon by codon comparison of dN-dS across the mammalian alignment of *BHLHE41*.** Positive dN-dS values represent positive selection, negative dN-dS values represent purifying selection, and zero dN-dS values represent neutrality. Codon # comes from the HyPhy output and does not include codons removed because >70% of sequences in the alignment had gaps (*e.g.*, the first 106 amino acids in gorilla). All the codons with significant p-values (red) have dN < dS. Blue points have dN-dS that are not significantly different from zero. There are no codons with significant dN > dS. Conserved domains in the *Homo sapiens BHLHE41* protein are indicated with black bars above the graph representing the codon positions for bHLH and the orange domain. Invariant codons are not shown because a p-value could not be calculated.

value of 0.99, while in the tree generated using maximum likelihood, these species have bootstrap values of 68%, which is just below the frequently used cut-off for reliability of 70% [33].

## Molecular evolution and variation around conserved domains

Among the species, there were no significant pairwise dN-dS values in the test for positive selection (all comparisons had p = 1.0). On the other hand, the Z-test for purifying selection revealed 96.8% of the pairwise species comparisons had dN-dS values significantly less than zero (S1 Table). The mean dN-dS value was -6.13 suggesting strong purifying selection.

After removing invariant codons and those with a gap in >70% of the sequences, we found 227 of 343 codons had significantly higher dS than dN values (66.2%) indicating strong purifying selection (S2 Table; Fig 3). The dN-dS value for the "short-sleeper" allele (P385R [10]) had a dN-dS value of -3.02 (p < 0.01), consistent with strong purifying selection. When compared to all 343 codons, P385R had the 45th most negative dN-dS value. Another variant known to affect sleep behavior in humans, Y362H [17], exhibited no variation in the codon and therefore no p-value could be calculated (S2 Table).

Although 3D structures are an integral part of determining a protein's function, there was no known 3D structure for *H. sapiens BHLHE41* protein. To confirm that there were no homologous sequences with known 3D structures in other species or sequences with alternative gene annotations, we conducted a BLAST search using the human *BHLHE41* sequence and filtered for results with known 3D structures. The best-hit had an E-value of 0.042, which is substantially above the commonly used threshold for homology ($<10^{-3}$; [34]), thus we conclude that no 3D structures for *BHLHE41* are currently available.

Instead, we compared variation in the two conserved domains from the *H. sapiens BHLHE41* protein GenBank flat file—the bHLH domain and the orange domain. There is no variation in the amino acid alignment across the 59 amino acids in the bHLH domain (S4

Table). The average dN-dS value for these codons is -0.891 suggesting purifying selection. All the p-values of variable codons in this region are <0.01 (S4 Table; Fig 3). The orange domain spans amino acids in the human coding sequence 129–175 (amino acids 235–281 in our alignment, S4 Table). There are two variable sites at human amino acid sites S147A (variants appear in pig, whales, dolphins, sheep and cow; dN-dS = 0.50) and P157Q (variants appear in leopard, northern fur seal and California sea lion; dN-dS = -0.141). The average dN-dS for the 47 codons in the orange domain is -0.811 (Fig 3). Of the 27 p-values that could be calculated in the orange domain, all but five have p-values <0.05. In general, there are large stretches of invariant amino acids among the mammalian samples (*e.g.*, residues 148 to 252 of our alignment). Furthermore, there are seven poly-alanine residues ranging from four to 16 amino acids in length between amino acid alignment positions 407 and 547.

## Discussion

### Strong purifying selection on *BHLHE41* in mammals

Through this study, we explored patterns of molecular evolution in the sleep-related, circadian clock gene *BHLHE41* in mammals. Overall, this gene is highly conserved among mammals consistent with its essential function. For example, the bHLH conserved domain shows no amino acid variation among mammals (and even the reptilian outgroup) (amino acid alignment positions 152–210 in S4 Table). Furthermore, the evolutionary history of this gene among mammals is consistent with well-established species-level phylogenetic relationships [19, 20].

In general, the consequences of purifying selection (aka background selection) have been described as "poorly understood [35]." We know that this type of selection arises when the rate of nonsynonmous substitutions (dN) is substantially lower than the rate of synonymous substitutions (dS) [18]. The difference in substitution rates occurs because most nonsynonymous substitutions in genes under purifying selection are deleterious and are removed from the population in order to preserve the biological function of the protein. Purifying selection explains amino acid sequence conservation across long evolutionary time periods [35]. Purifying selection by definition reduces genetic diversity at both the codons under direct selection and those linked to codons under selection [35]. Genes under purifying selection tend to be essential for biological function, highly expressed, and employed in vital developmental pathways [36] like sleep regulation. The strong footprint of purifying selection that we detected in the sleep related gene, *BHLHE41*, is consistent with its essential role in sleep regulation [5]. Yet, the expectations under purifying selection lie in stark contrast with observed non-synonymous substitutions recorded in humans [10,17] that originally inspired this study (*i.e.* "short sleeper allele). For example, at least two nonsynonymous substitutions found in humans (P385R and Y362H) are not lethal and in fact confer altered sleep patterns that may even be adaptive under certain circumstances [10,17].

### Unexpected length variation in gorilla *BHLHE41*

Unexpectedly, we found two large indels in the gorilla homologue for *BHLHE41*. The 318 base pair insertion at the 5' end of the coding sequence suggests a start codon 106 amino acids upstream from the remaining mammalian start codons. It is noteworthy that the gorilla sequence still contains AUG at the site where the remaining sequences start translation. Additionally, the gorilla sequence contains a 195 base pair deletion near the 3' end of the coding sequence. This predicted deletion includes both short-sleeper variants previously described (Y362H and P385R)—essential amino acids for proper circadian clock function [10, 14]. Although these indels may reflect novel function of *BHLHE41* in gorilla, these animals are not

known to have particularly unusual sleep patterns, nor disrupted circadian clocks as would be expected from the addition of 106 amino acids on the 5' end and the deletion of 65 amino acids from near the 3' end.

The existence of these indels seems especially unlikely given the widespread pattern of purifying selection on this gene across mammals (>97% of pairwise species comparisons) and across codons (~50% of codons). The 5' insertion is especially suspicious because it is unique among the 27 mammal sequences investigated and without it, the sequence aligns perfectly with the rest of the mammalian start codons. Although this insertion does not immediately affect the bHLH conserved domain, such a large insertion within 50 residues seems very likely to disrupt protein folding in this region. Without a known 3D structure, confident determination of the effects of these indels on the 3D structure and therefore function remain unknown.

The gorilla *BHLHE41* sequence was produced during whole-genome sequencing and was predicted using an annotation pipeline [32]. There is no literature discussing this unusual gorilla *BHLHE41* sequence. Unfortunately, there is no cDNA sequence for this gene from gorilla in Genbank release 233.0 (April 2019). Therefore, we suggest that there may have been an error in the identification of the start codon by the open reading frame search algorithm [37].

An error in the open reading frame detection algorithm may account for the incorrectly identified start codon. It is noteworthy that He et al. [10] suggested only four introns, yet EMBL identified five introns. Furthermore, EMBL indicates this gorilla amino acid sequence is only 419aa long compared to the Genbank accession which measures 522 residues (S3 Fig). Experimental determination of the length of the gorilla *BHLHE41* protein by sequencing cDNA or RNA-Seq will be necessary in order to determine the true start codon in gorilla (or start codons if there are multiple isoforms of this gene) and the validity of the 195bp deletion near the 3' end of the coding sequence.

There is no evidence of alternative splice variants for *BHLHE41* in Gorilla according to EMBL (ENSGGOG00000015498; accessed June 13, 2019). Furthermore, although there are 11 paralogues in EMBL, all are less than 37% identical to *BHLHE41* indicating significant sequence divergence and unlikely to be mistaken for *BHLHE41*. If these were paralogous sequences, they would most likely show incongruent relationships with the well-established mammalian phylogeny. The status in the UNIPROTKB database indicates it is still only a predicted protein with an Annotation Score of 2/5 (G3RHJ7_GORGO). It is noteworthy that the EMBL transcript protein sequence contains neither the early start codon, nor the 195bp deletion in the coding sequence (ENSGGOT00000015550.3). However, the Genbank Annotation Release 101 of the *Gorilla gorilla gorilla* genome (Nov 4 2016) still contains these two large indels. A very recent, new genome sequence of a different gorilla individual (Kamilah, GCA_000151905.3, Aug. 28, 2019) no longer exhibits the 195bp deletion near the 3' end of the coding sequence. No annotations were available for this genome, but hopefully it eventually includes a start codon that matches the rest of mammals.

The annotation of protein-coding genes is currently based on gene prediction algorithms [37]. Gene prediction algorithms have been through several revolutions since their initial application [38,39]. Majoros et al. [40] evaluated the quality of gene prediction algorithms. An evaluation of gene finders based on hidden markov models (HMMs) was done by Knapp & Chen [41]; the authors reported that no significant improvement in the quality of de novo gene prediction methods occurred during the previous 5 years. Bakke et al. [42] evaluated three second-generation gene annotation systems on the genome of the archaeon *Halorhabdus utahensis* from the performance of the gene-prediction models to the functional assignments of genes and pathways. Comparison of gene-calling methods showed that 90% of all three annotations share exact stop sites with the other annotations, but only 48% of identified genes share both start and stop sites [42]. Palleja et al. [43] performed an interesting investigation of

overlapping CDS in prokaryotic genomes. They compared overlapping genes with their corresponding orthologues and found that more than 900 reported overlaps larger than 60 bp were not real overlaps, but annotation errors. Given that *BHLHE41* is just one of the 46,653 coding sequences predicted in gorilla, we are cautious about making any widespread conclusions about the remaining loci.

To avoid annotation mistakes, Armengaud [44] recommends using proteomics in association with translations in all six reading frames. Prasad et al. [38] provide a method combining transcriptome and proteomics to aid in genome annotation. However, genes that are expressed only under special conditions or in rarely sampled tissues, or whose expression is below the detection level, pose a challenge even for proteomic and cDNA validation.

## Conclusions

We sought to determine if there was a footprint of positive selection on *BHLHE41* in mammals in light of its effect on sleep behaviors. We found that the majority of the *BHLHE41* coding sequences exhibit a history of purifying selection (especially the conserved domains), indicating the gene has an essential function for survival and reproduction. In particular, if adaptive sleep behaviors are conferred by *BHLHE41*, we predicted residues 362 and/or 385 to show a history of positive selection. Both sites were invariant across mammals consistent with strong purifying selection on the underlying codons. The evolutionary history of *BHLHE41* is largely congruent with the well-established mammalian phylogeny indicative of homologous comparisons. From the single sequences we used per species for a limited number of mammals, we found no other species (besides humans) that exhibited the two "short-sleeper" variants [10]. These sites are likely undergoing strong purifying selection in most mammalian species. Additional population-level sampling across a broader diversity of mammals would be required to accurately determine if these variants are truly unique to humans. During our investigation, we discovered an unusually annotated sequence for *G. gorilla gorilla*. We suggest that the early start codon and deletion near the 3' end are annotation errors that warrant experimental verification.

## Supporting information

**S1 Fig. Verification that the Bayesian MCMC phylogenetic search reached stationarity.**
(DOCX)

**S2 Fig. Phylogenetic tree of euteleostomi *BHLHE41* mRNA using Bayesian analysis.**
(DOCX)

**S3 Fig. EMBL structure of the transcript for *BHLHE41* from *gorilla gorilla gorilla* with conserved domains indicated.**
(DOCX)

**S1 Table. Pairwise codon-based test of purifying selection for mammalian *BHLHE41*.**
(DOCX)

**S2 Table. Codon-by-codon test for selection.**
(DOCX)

**S3 Table. *BHLHE41* mammalian nucleotide alignment with reptile outgroup.**
(DOCX)

**S4 Table. *BHLHE41* mammalian amino acid alignment with reptile outgroup.**
(DOCX)

## Acknowledgments

The authors thank Santa Clara University's Department of Biology for providing access to the computer lab running Geneious software, especially Daryn Baker and Steve Hines. We acknowledge the students in BIOL178 who provided valuable feedback during the early stages of this research. Aleezah Salmaan provided valuable editorial advice on an earlier draft of this study.

## Author Contributions

**Conceptualization:** Krishna Unadkat, Justen B. Whittall.

**Investigation:** Krishna Unadkat, Justen B. Whittall.

**Writing – original draft:** Krishna Unadkat.

**Writing – review & editing:** Krishna Unadkat, Justen B. Whittall.

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
