## [Decision Letter · Decision Letter 0]

17 Dec 2019

PONE-D-19-25843

Unexpected predicted length variation for the coding sequence of the sleep related gene, BHLHE41 in gorilla amidst strong purifying selection across mammals

PLOS ONE

Dear Dr. Whittall,

Thank you for submitting your manuscript to PLOS ONE. After careful consideration, we feel that it has merit but does not fully meet PLOS ONE’s publication criteria as it currently stands. Therefore, we invite you to submit a revised version of the manuscript that addresses the points raised during the review process.

Reviewer #2 makes a number of very good suggestions, which I invite you to follow. While PLOS One does not have any criteria of significance, it is important that the presentation of the article be consistent with the results, as suggested. It is also important to compare fairly species with or without population data, especially in the discussion. Finally, please do use the new Gorilla genome reference sequence.

We would appreciate receiving your revised manuscript by Jan 31 2020 11:59PM. To enhance the reproducibility of your results, we recommend that if applicable you deposit your laboratory protocols in protocols.io, where a protocol can be assigned its own identifier (DOI) such that it can be cited independently in the future. For instructions see: http://journals.plos.org/plosone/s/submission-guidelines#loc-laboratory-protocols

We look forward to receiving your revised manuscript.

Kind regards,

Marc Robinson-Rechavi

Academic Editor

PLOS ONE

Journal Requirements:

Please ensure that your manuscript meets PLOS ONE's style requirements, including those for file naming. The PLOS ONE style templates can be found at http://www.plosone.org/attachments/PLOSOne_formatting_sample_main_body.pdf and http://www.plosone.org/attachments/PLOSOne_formatting_sample_title_authors_affiliations.pdf

Reviewers' comments:

Reviewer's Responses to Questions

**Comments to the Author**

1. Is the manuscript technically sound, and do the data support the conclusions?

Reviewer #1: Yes

Reviewer #2: Partly

2. Has the statistical analysis been performed appropriately and rigorously? 

Reviewer #1: Yes

Reviewer #2: Yes

3. Have the authors made all data underlying the findings in their manuscript fully available?

Reviewer #1: Yes

Reviewer #2: Yes

4. Is the manuscript presented in an intelligible fashion and written in standard English?

Reviewer #1: Yes

Reviewer #2: No

5. Review Comments to the Author

Reviewer #1: In this study, BHLH41 genes related with circadian clock was aligned and compared in 27 mammals. The result shows that the gorilla BHLHE 41 sequence has indels which were not found in other mammals. However, this variation in sequence was not verified. Overall, this study is superficial, no much information can we get from this study. The only interesting point about gorilla BLHLH variation isn’t supported by experimental validation as well.

Reviewer #2: Summary

In this manuscript, the authors investigated the evolutionary history of a circadian clock gene BHLHE41 in mammals. In humans, this gene is known to harbor two variants linked to a short-sleeper behavior. The authors compared homologous nucleotide and protein sequences to the human BHLHE41 sequences for 27 mammals and 1 reptilian outgroup. They performed phylogenetic reconstruction of the gene evolutionary history in these species and showed it was consistent with the species phylogeny. Using sequence alignments for the 27 mammals, they showed that most of the gene was under purifying selection, with no codon under positive selection. Finally, they found a 318bp insertion at the beginning of BHLHE41 sequence and a 195bp deletion near the end of the sequence in Gorilla gorilla gorilla, which they suggest to be errors from the automatic annotation process in this species.

Comments

Overall, I think the analyses performed in this study are technically correct and yielded results that could be of interest to other researchers in the field. However, I have several concerns about the manuscript’s structure and the way results are presented. I also have minor comments about several aspects of the manuscripts, which will be listed after the main concerns.

- To me, the manuscript is not very clear in its current form. This could be addressed by better defining the aims of the study, namely which results is the manuscript focusing on. At the moment, there are two main results presented in the paper: 1) conservation of BHLHE41 among mammals and no sign of selection for short-sleeper variants, and 2) potential annotation errors in Gorilla gorilla gorilla from the automated annotation pipeline. The introduction mostly focuses on analyses related to the first results, while the discussion is almost entirely dedicated to the second result. I personally think that the study of conservation of BHLHE41’s sequence among mammals is of higher interest to the community than the annotation error, which is not present in ENSEMBL’s annotation, and I suggest that the paper be reorganized so that the different sections are more homogeneous and the logical flow is easier to follow overall.

- If the authors would like to keep the discovery of uncharacteristic indels in G. gorilla gorilla as a main result, I suggest dedicating a clear section in the results to the analyses supporting their claims. I think this section should include some of the analyses mentioned in the discussion, e.g. the analysis of G. gorilla gorilla genome and the comparison with the ENSEMBL annotation.

- A new Gorilla gorilla gorilla genome is available since 2019/08/28 and is now the reference genome for this species on NCBI Genbank/RefSeq. This genome is briefly mentioned in the discussion, but the manuscript was not updated accordingly. In particular, accession numbers and genomic positions have changed for any Gorilla sequence, including chr12 in the discussion. The accession number for gorilla BHLHE41 is now XM_031000846.1 (XM_019037881 does not yield any results). Fortunately, it seems the sequence of BHLHE41 has not changed at all, but this should be double-checked. Other analyses involving the Gorilla genome (e.g. discussion) should be updated with this new assembly.

- In the conclusion, page 21, the authors claim “From the available mammalian sequences, it appears that the “short-sleeper” variant is only present in humans”. I do not think you can reach this conclusion without population data from other species; the variants are also absent from the human reference genome.

- Overall, I strongly suggest that the manuscript be revised for language and structure within each section. The manuscript is overall well structured, but the organization within each section is not always easy to follow and lacks a clear logical flow.

In addition to these concerns, I have several minor comments:

- I think the results section of the abstract is too detailed. I would recommend simplifying the summary of results to make the abstract more engaging to the reader.

- There are several issues with formatting (things like e.g. and i.e. should be in italic)

- Some sections of the introduction seem outside the scope of a research article, e.g. explaining dN/dS. In the introduction, I would suggest talking about “estimating selection” rather than “comparing dN and dS”.

- The last paragraph of the introduction is not very clear; I expected a clearer “aims, methods, results” structure to facilitate reading the rest of the manuscript.

Methods:

- Why did you blast the human BHLHE41 sequence instead of using already existing annotation? All the sequences you found were annotated as BHLHE41 already.

- Why did you restrict the comparison to these species? Searching “(BHLHE41) AND "mammals"[porgn:__txid40674]” in the proteins database on NCBI yields 165 results. There may be a good reason for choosing these 28 species but then it should be clearly stated.

Results:

- Querying sequence and BLAST are not results in my opinion. I think they would fit better in the methods section as a description of the sequences used for the comparisons.

- It is not clear to me what the phylogenetic analyses bring to the paper. In the discussion, these results are used to justify that the history of the gene follows that of the species, but this result is not really connected to the rest of the paper. I think it could be moved to supplementary information, or better integrated in the study.

- In the section “Molecular Evolution and Variation around Conserved Domains”, page 16: the sentence “Additionally, a BLAST search revealed there were no sequences that had known protein structures in NCBI’s protein data bank with E-values below 0.042, which is above the commonly used threshold for homology (<10-3 ; [33])” is not clear. Do you mean there were no sequences homologous to that of BHLHE41 with known protein structure?

- The legend for Figure 1 is not very clear.

- When referring to the Gorilla genome (in the discussion at the moment), give the accession number of the assembly (GCA_000151905.3) instead of the bioproject.

- If both names refer to the same domain, I think it would be better to use consistent names, e.g. bHLH vs HLH

- In the discussion, pages 18-19: “We searched the Gorilla gorilla gorilla chromosome 12 whole genome shotgun sequence (NC_018436) between bp 58,885,949 and 58,889,015 and found that although the unusual 318bp 19 upstream from the mammalian start codon exists, the gorilla annotation actually identified the correct start codon (no 318bp insertion on the 5’ end)”. If the gorilla annotation is different from the predicted mRNA, it should be stated clearly.

6. PLOS authors have the option to publish the peer review history of their article (what does this mean?). If published, this will include your full peer review and any attached files.

Reviewer #1: No

Reviewer #2: No

---

## [Author Response · Author response to Decision Letter 0]

9 Mar 2020

March 9, 2020

Dear PLOS ONE Editor,

Thank you for sharing the two reviews of our manuscript (PONE-D-19-25843) entitled, “Unexpected predicted length variation for the coding sequence of the sleep related gene, BHLHE41 in gorilla amidst strong purifying selection across mammals.” Below you will find our responses to your comments and both reviewers’ comments in bold font. Page and line numbers refer to those in the WORD version of this revised manuscript.

Editor’s Comments extracted from decision email: 

Reviewer #2 makes a number of very good suggestions, which I invite you to follow. 

We agree. See our responses below.

While PLOS One does not have any criteria of significance, it is important that the presentation of the article be consistent with the results, as suggested. 

We have made substantial changes to re-align the paper with the results spanning the Abstract, Introduction, Results, Discussion and References sections. Our specific changes have been itemized in our response to Reviewer #2 below.

It is also important to compare fairly species with or without population data, especially in the discussion. 

We agree that population-level data might be necessary to determine variation within a species. However, we chose to sample broadly across mammals (27 species) to look for species-level changes or above. Unfortunately, due to limited population-level data available for most of these species, we were unable to conduct thorough searches for variants for these diverse mammal species. Therefore, we have qualified our results based on the sampling in the revised Discussion. 

Finally, please do use the new Gorilla genome reference sequence.

Done & updated, but this did not change the results.

Reviewer #1: In this study, BHLH41 genes related with circadian clock was aligned and compared in 27 mammals. The result shows that the gorilla BHLHE 41 sequence has indels which were not found in other mammals. However, this variation in sequence was not verified. Overall, this study is superficial, no much information can we get from this study. The only interesting point about gorilla BLHLH variation isn’t supported by experimental validation as well.

We thank the reviewer for identifying the gorilla BHLH variation as an “interesting point”. This finding is now highlighted with a unique subheader within the Results section entitled, “Sequence Length Variation in Gorilla” (P13, L322). In regards to the reviewer’s concern that the study is “superficial” and doesn’t have much to offer, I would point the reviewer to the seven “Criteria for Publication” at PLoS ONE, none of which suggest anything about the general interest of the contribution. My understanding of the PLoS ONE philosophy is to publish manuscripts that satisfy the criteria for publication (rigorous and responsible science) and then let the readership determine whether it is “superficial” and how “interesting” it really is. We note that bioinformatics is an established field that does not require “experimental validation”. In fact, this report will hopefully stimulate such follow up study once this is part of the scientific record. Finally, I point to Reviewer #2 who clearly indicated that the “analyses performed in this study are technically correct and yielded results that could be of interest to other researchers in the field”.

Reviewer #2: Summary

In this manuscript, the authors investigated the evolutionary history of a circadian clock gene BHLHE41 in mammals. In humans, this gene is known to harbor two variants linked to a short-sleeper behavior. The authors compared homologous nucleotide and protein sequences to the human BHLHE41 sequences for 27 mammals and 1 reptilian outgroup. They performed phylogenetic reconstruction of the gene evolutionary history in these species and showed it was consistent with the species phylogeny. Using sequence alignments for the 27 mammals, they showed that most of the gene was under purifying selection, with no codon under positive selection. Finally, they found a 318bp insertion at the beginning of BHLHE41 sequence and a 195bp deletion near the end of the sequence in Gorilla gorilla gorilla, which they suggest to be errors from the automatic annotation process in this species.

Comments

Overall, I think the analyses performed in this study are technically correct and yielded results that could be of interest to other researchers in the field. However, I have several concerns about the manuscript’s structure and the way results are presented. I also have minor comments about several aspects of the manuscripts, which will be listed after the main concerns.

We would like to thank Reviewer #2 for taking the time to carefully read, assess and make thoughtful, constructive comments for improving the manuscript. Few Reviewers would go so far as to repeat our initial BLAST to identify that there were more than 27 mammalian sequences available. We appreciate this and have integrated nearly every one of their suggestions.

- To me, the manuscript is not very clear in its current form. This could be addressed by better defining the aims of the study, namely which results is the manuscript focusing on. 

We have clarified the aims by restructuring the last paragraph of the Introduction (P1, L121-125). It now states, “There are two clear aims of this study…”. Of course, during our investigation we discovered an unexpected length difference which wasn’t one of the original aims, but we felt deserved reporting and discussing later in the manuscript (see changes made in this regard in our responses below).

At the moment, there are two main results presented in the paper: 1) conservation of BHLHE41 among mammals and no sign of selection for short-sleeper variants, and 2) potential annotation errors in Gorilla gorilla gorilla from the automated annotation pipeline. The introduction mostly focuses on analyses related to the first results, while the discussion is almost entirely dedicated to the second result. I personally think that the study of conservation of BHLHE41’s sequence among mammals is of higher interest to the community than the annotation error, which is not present in ENSEMBL’s annotation…

We thank the Reviewer for suggesting that the sequence evolution is of “higher interest” than the length variation in the Gorilla sequence (although not one of the Criteria for Publication as mentioned in our response to Reviewer #1). We agree and have kept the Introduction focused on the sequence variation and expanded the Discussion of the remarkable sequence conservation across mammals. In fact, in the original draft, there were seven paragraphs discussing the possibility of an annotation error in the gorilla genome, while at the same time there was only one paragraph dedicated to the discussion of purifying selection discovered in the study. 

As suggested by the reviewer #2, we have further developed the Discussion by giving it a subheading “Strong Purifying Selection on BHLHE41 in Mammals” and elaborating on our purifying selection discovery across mammals therein. This includes an additional paragraph in the Discussion and additional references that align with our specific results (P18, L532-548).

I suggest that the paper be reorganized so that the different sections are more homogeneous and the logical flow is easier to follow overall.

Agreed. Per the Reviewer #2’s comment, we have added two descriptive headers to the Discussion in order to maintain a clear distinction between the two main findings of our study. The new headers are “Strong Purifying Selection on BHLHE41 in Mammals” and “Unexpected Length Variation in Gorilla BHLHE41”. 

- If the authors would like to keep the discovery of uncharacteristic indels in G. gorilla gorilla as a main result, I suggest dedicating a clear section in the results to the analyses supporting their claims. I think this section should include some of the analyses mentioned in the discussion, e.g. the analysis of G. gorilla gorilla genome and the comparison with the ENSEMBL annotation.

Great idea! We moved half of the third paragraph in the Discussion regarding length variation in the Gorilla gene and appended it as a second paragraph to a newly dedicated section in the Results called “Sequence Length Variation in Gorilla.” I think this helps structure the Results and the Discussion.

- A new Gorilla gorilla gorilla genome is available since 2019/08/28 and is now the reference genome for this species on NCBI Genbank/RefSeq. This genome is briefly mentioned in the discussion, but the manuscript was not updated accordingly. In particular, accession numbers and genomic positions have changed for any Gorilla sequence, including chr12 in the discussion. The accession number for gorilla BHLHE41 is now XM_031000846.1 (XM_019037881 does not yield any results). Fortunately, it seems the sequence of BHLHE41 has not changed at all, but this should be double-checked. Other analyses involving the Gorilla genome (e.g. discussion) should be updated with this new assembly.

Thank you for informing us about this update. We have replaced the old IDs with the new ones in Table 1 and in the “Sequence Length Variation in Gorilla” section (P13, L278).

- In the conclusion, page 21, the authors claim “From the available mammalian sequences, it appears that the “short-sleeper” variant is only present in humans”. I do not think you can reach this conclusion without population data from other species; the variants are also absent from the human reference genome.

We have qualified our conclusion in light of our limited sampling (one sequence per species for only 27 mammal species). This conclusion sentence now reads, “From the single sequences we used per species for a limited number of mammals, we found no other species (besides humans) that exhibits the “short-sleeper” variants…” (P22, L642-648). This is followed by the suggestion for future research in the following sentence, “Additional population-level sampling across a broader diversity of mammals would be required to accurately determine if these variants are truly unique to humans.” (P22, L649-650).

- Overall, I strongly suggest that the manuscript be revised for language and structure within each section. The manuscript is overall well structured, but the organization within each section is not always easy to follow and lacks a clear logical flow.

We appreciate the suggestion and made numerous small restructuring edits throughout the manuscript as can be seen from the track changes version.

In addition to these concerns, I have several minor comments:

- I think the results section of the abstract is too detailed. I would recommend simplifying the summary of results to make the abstract more engaging to the reader.

Done.

- There are several issues with formatting (things like e.g. and i.e. should be in italic)

Fixed what we could find.

- Some sections of the introduction seem outside the scope of a research article, e.g. explaining dN/dS. In the introduction, I would suggest talking about “estimating selection” rather than “comparing dN and dS”.

The 3rd to last paragraph of the Introduction that the reviewer is referring to starts with a description of selection (as the reviewer recommends). We don’t mention dN-dS until the end of the 3rd sentence and in that case, only to provide the necessary background for the reader to later interpret our results which are reported in those terms (e.g., Fig 3 and Table S2). Therefore, we have not changed this specific section. However, we did tighten up the Introduction in several other paragraphs where the background material was superfluous (like 2 general sentences in the first paragraph of the Intro; P3, L57-58).

- The last paragraph of the introduction is not very clear; I expected a clearer “aims, methods, results” structure to facilitate reading the rest of the manuscript.

Thank you for pointing this out. We agree that this “signpost” paragraph is essential to cue readers for what’s to come. We now present two clear “Aims” and a brief description of our Methods hinting at the type of Results we will present (P6, L137-146).

Methods:

- Why did you blast the human BHLHE41 sequence instead of using already existing annotation? All the sequences you found were annotated as BHLHE41 already.

Great question. We did this intentionally. Using BLAST ensures we find all sequences with significant similarity regardless of annotation errors. If we had only relied on the annotations, we could accidentally pickup sequences that were mistakenly named BHLHE41 or more likely, we would have missed sequences that were annotated with a different name (or parts of genomes that remain unannotated). As we mention in the Introduction, this gene is also known as DEC2 (2nd paragraph) so using BLAST casts a wider net since it relies on the sequence content, not the annotation that can sometimes come with ambiguity or even mistakes.

- Why did you restrict the comparison to these species? Searching “(BHLHE41) AND "mammals"[porgn:__txid40674]” in the proteins database on NCBI yields 165 results. There may be a good reason for choosing these 28 species but then it should be clearly stated.

Yes, there are more sequences, but many are partial coding sequences that we chose not to include for thoroughness of the sequence evolution analysis. Furthermore, many of those additional sequences are identical sequences from the same species. After a preliminary analysis of all full-length CDS (including multiple sequences for some species), we confirmed that those duplicates were not informative. Therefore, we arbitrarily selected one sequence per species and that is how we arrived at the 27 mammalian sequences included herein.

Results:

- Querying sequence and BLAST are not results in my opinion. I think they would fit better in the methods section as a description of the sequences used for the comparisons.

Good suggestion. Moved (P7, L167-162).

- It is not clear to me what the phylogenetic analyses bring to the paper. In the discussion, these results are used to justify that the history of the gene follows that of the species, but this result is not really connected to the rest of the paper. I think it could be moved to supplementary information, or better integrated in the study.

Thank you for this feedback. We feel it is essential and have attempted to justify its placement in the manuscript. Briefly, in order to test for selection on the genes and individual codons, we have to compare orthologues (not gene duplicates = paralogues, since they often show different patterns of dN/dS following silencing or neo-functionalization). For an individual gene in mammals, we can assess orthology using phylogenetic analysis since most relationships are well established in this lineage (for example, see Kemp’s The Origin and Evolution of Mammals from 2005). 

We have updated the phylogenetic analysis with the following justification in the Methods section entitled, “Phylogenetic Analysis”. It reads, “In order to test for homology and confirm that we were comparing orthologous sequences, we conducted maximum likelihood and Bayesian phylogenetic analyses. If the evolutionary relationships of the BHLHE41 coding sequence reflects the known relationships among mammals, then we can conclude homology and proceed with the tests for selection.” (P8, L205-208).

- In the section “Molecular Evolution and Variation around Conserved Domains”, page 16: the sentence “Additionally, a BLAST search revealed there were no sequences that had known protein structures in NCBI’s protein data bank with E-values below 0.042, which is above the commonly used threshold for homology (<10-3 ; [33])” is not clear. Do you mean there were no sequences homologous to that of BHLHE41 with known protein structure?

Yes, there are no homologous sequences with 3D structures. We have clarified this Result.

- The legend for Figure 1 is not very clear.

Thank you for the feedback. We reordered the description of the figure to start with the explanation of the overall aspects (sequence identity graph and the location of the short sleeper variants as arrows) and then listed the two unexpected findings in Gorilla. 

- When referring to the Gorilla genome (in the discussion at the moment), give the accession number of the assembly (GCA_000151905.3) instead of the bioproject.

Done.

- If both names refer to the same domain, I think it would be better to use consistent names, e.g. bHLH vs HLH

Done, we have changed to bHLH throughout.

- In the discussion, pages 18-19: “We searched the Gorilla gorilla gorilla chromosome 12 whole genome shotgun sequence (NC_018436) between bp 58,885,949 and 58,889,015 and found that although the unusual 318bp 19 upstream from the mammalian start codon exists, the gorilla annotation actually identified the correct start codon (no 318bp insertion on the 5’ end)”. If the gorilla annotation is different from the predicted mRNA, it should be stated clearly.

Done.

---

## [Decision Letter · Decision Letter 1]

26 Mar 2020

Unexpected predicted length variation for the coding sequence of the sleep related gene, BHLHE41 in gorilla amidst strong purifying selection across mammals

PONE-D-19-25843R1

Dear Dr. Whittall,

We are pleased to inform you that your manuscript has been judged scientifically suitable for publication and will be formally accepted for publication once it complies with all outstanding technical requirements.

With kind regards,

Marc Robinson-Rechavi

Academic Editor

PLOS ONE

Additional Editor Comments (optional):

Reviewers' comments:

Reviewer's Responses to Questions

**Comments to the Author**

1. If the authors have adequately addressed your comments raised in a previous round of review and you feel that this manuscript is now acceptable for publication, you may indicate that here to bypass the “Comments to the Author” section, enter your conflict of interest statement in the “Confidential to Editor” section, and submit your "Accept" recommendation.

Reviewer #2: All comments have been addressed

2. Is the manuscript technically sound, and do the data support the conclusions?

Reviewer #2: Yes

3. Has the statistical analysis been performed appropriately and rigorously? 

Reviewer #2: Yes

4. Have the authors made all data underlying the findings in their manuscript fully available?

Reviewer #2: Yes

5. Is the manuscript presented in an intelligible fashion and written in standard English?

Reviewer #2: Yes

6. Review Comments to the Author

Reviewer #2: I would like to thank the authors for considering and implementing my suggestions and for answering in a constructive manner. I also appreciate that the authors have made extra changes following the "spirit" of the comments, and not just answered point by point. I think the manuscript is now suitable for publication.

7. PLOS authors have the option to publish the peer review history of their article (what does this mean?). If published, this will include your full peer review and any attached files.

Reviewer #2: Yes: Romain Feron

---

## [Editor Report · Acceptance letter]

30 Mar 2020

PONE-D-19-25843R1 

Unexpected predicted length variation for the coding sequence of the sleep related gene, *BHLHE41* in gorilla amidst strong purifying selection across mammals 

Dear Dr. Whittall:

I am pleased to inform you that your manuscript has been deemed suitable for publication in PLOS ONE. Congratulations! Your manuscript is now with our production department. 

With kind regards,

on behalf of

Prof. Marc Robinson-Rechavi 

Academic Editor

PLOS ONE